# Comparison of Machine Learning Models and the Fatty Liver Index in Predicting Lean Fatty Liver

**DOI:** 10.3390/diagnostics13081407

**Published:** 2023-04-13

**Authors:** Pei-Yuan Su, Yang-Yuan Chen, Chun-Yu Lin, Wei-Wen Su, Siou-Ping Huang, Hsu-Heng Yen

**Affiliations:** 1Department of Internal Medicine, Division of Gastroenterology, Changhua Christian Hospital, Changhua 500, Taiwan; 27716@cch.org.tw (Y.-Y.C.); 35301@cch.org.tw (W.-W.S.); 182972@cch.org.tw (S.-P.H.); 91646@cch.org.tw (H.-H.Y.); 2College of Medicine, National Chung Hsing University, Taichung 400, Taiwan; 3Department of Hospitality Management, MingDao University, Changhua 500, Taiwan; 4Department of Family Medicine, Yumin Hospital, Nantou 540, Taiwan; amonslin@gmail.com; 5General Education Center, Chienkuo Technology University, Changhua 500, Taiwan; 6Department of Electrical Engineering, Chung Yuan Christian University, Taoyuan 320, Taiwan; 7Artificial Intelligence Development Center, Changhua Christian Hospital, Changhua 500, Taiwan

**Keywords:** lean fatty liver, machine learning model, fatty liver index

## Abstract

The reported prevalence of non-alcoholic fatty liver disease in studies of lean individuals ranges from 7.6% to 19.3%. The aim of the study was to develop machine-learning models for the prediction of fatty liver disease in lean individuals. The present retrospective study included 12,191 lean subjects with a body mass index < 23 kg/m^2^ who had undergone a health checkup from January 2009 to January 2019. Participants were divided into a training (70%, 8533 subjects) and a testing group (30%, 3568 subjects). A total of 27 clinical features were analyzed, except for medical history and history of alcohol or tobacco consumption. Among the 12,191 lean individuals included in the present study, 741 (6.1%) had fatty liver. The machine learning model comprising a two-class neural network using 10 features had the highest area under the receiver operating characteristic curve (AUROC) value (0.885) among all other algorithms. When applied to the testing group, we found the two-class neural network exhibited a slightly higher AUROC value for predicting fatty liver (0.868, 0.841–0.894) compared to the fatty liver index (FLI; 0.852, 0.824–0.81). In conclusion, the two-class neural network had greater predictive value for fatty liver than the FLI in lean individuals.

## 1. Introduction

Non-alcoholic fatty liver disease (NAFLD) is one of the most prevalent forms of liver disease worldwide and is associated with increased risks of cirrhosis and hepatocellular carcinoma (HCC) [1]. Incidence rates of HCC have been higher in patients with nonalcoholic steatohepatitis (NASH) cirrhosis than in those with non-cirrhotic NAFLD. In Asian populations, a Japanese study revealed a 5-year HCC incidence of 11.3% among patients with NASH cirrhosis [2]. The global prevalence of NAFLD has been reported as 25.24%, with a regional prevalence of 27.37% in Asia [3]. Fatty liver in lean individuals has similar clinical characteristics to NAFLD [4]. In fact, a Swedish registry study found that although lean individuals with NAFLD had a lower baseline fibrosis severity than did non-lean individuals, they were still at high risk for developing severe liver disease [5]. Lean individuals with NAFLD are at increased risk of developing type 2 diabetes mellitus (DM) and metabolic syndrome and increased risk of mortality from cardiovascular and liver diseases [4,6]. Lean and non-lean individuals with NAFLD share several metabolic risk factors, including hypertriglyceridemia, low high-density lipoprotein (HDL) levels, type 2 DM, hypertension, metabolic syndrome, increased body mass index (BMI) and increased waist circumference. Generic characteristics, such as the PNPLA3 G allele, also play an important role in the development of NAFLD among lean individuals. The prevalence of NAFLD in lean individuals according to different BMI criteria (23–25 kg/m^2^) reportedly ranges from 7.6% to 19.3% [7]. The prevalence of NAFLD in a population with a BMI < 24 kg/m^2^ from south Taiwan was reported as 18.5% [8].

NAFLD can be diagnosed by liver biopsy or image. The gold standard method for the diagnosis of steatosis is liver biopsy; however, this approach is invasive and carries some bleeding risk. Meanwhile, diagnostic imaging, such as ultrasonography, computed tomography and magnetic resonance imaging, is time-consuming, costly and not always available in remote areas. Early diagnosis of hepatic steatosis based on risk factors helps clinicians identify the adverse events of NAFLD and prescribe more lifestyle interventions to prevent them. In addition, the diagnosis of NAFLD in lean individuals can easily be missed. Several biomarkers have been investigated as predictors of fatty liver, including markers of apoptosis and oxidative stress, the BARD score and the fatty liver index (FLI) [9]. The FLI is the most validated tool for predicting hepatic steatosis in the general and lean population [10]. Artificial intelligence (AI) has also been used to predict NAFLD for several years. Electronic health records and imaging data are the two main sources of data used to develop machine-learning models [11]. The area under the receiver operating characteristic (AUROC) for predicting NAFLD was higher for AI-assisted ultrasound (0.98) compared to AI-assisted clinical data sets (0.85). Our previous report demonstrated that a machine learning model using extreme gradient boosting (XGBoost) had a greater predictive value for fatty liver [12]. However, there was limited data on using machine learning models to predict lean fatty liver in the world. The aim of our study is trying to develop machine learning models for the prediction of lean fatty liver and to compare these models with FLI in the lean population.

## 2. Materials and Methods

### 2.1. Patient Selection

This retrospective study included subjects that had received a health checkup at Changhua Christian Hospital between January 2009 and January 2019. All subjects were adults (20–80 years old) and “lean,” as defined by a BMI < 23 kg/m^2^. Collected clinical data and ultrasound findings were from the same day. Only subjects with complete data for all parameters, including complete blood counts and biochemistry and lipid profiles, were included in the study analysis. We excluded subjects with incomplete clinical parameters and those whose BMI was ≥23 kg/m^2^. Initially, 45,006 subjects were enrolled. We then excluded 13,076 subjects with incomplete clinical datasets and 19,739 subjects whose BMI was ≥ 23 kg/m^2^. Finally, a total of 12,191 subjects were included in the study (Figure 1). The present study was approved by the Ethics Committee of Changhua Christian Hospital (CCH IRB No: 191012), and informed consent was waived as all data were deidentified.

### 2.2. Ultrasound Imaging

Fatty liver disease was defined as moderate-to-severe fatty change on ultrasonography. All participants fasted for at least 6 h prior to ultrasound examinations. Ultrasonography was performed by three independent ultrasound operators who were blinded to clinical data. Moderate-to-severe fatty changes on ultrasound were defined as the presence of at least 3 of the following 4 features: (1) hepatorenal echo contrast, (2) liver brightness, (3) deep attenuation and (4) vascular blurring [13,14].

### 2.3. Model Construction and Validation

The dataset was divided into a training group (70%, 8533 subjects) and a testing group (30%, 3568 subjects) using a randomized 70–30 split. Due to the low prevalence of fatty liver among the lean individuals included in the present study, healthy controls and participants with fatty liver disease were randomized into each group using a 2:1 ratio (Figure 1). The training set and validation set were randomly selected from the training group using an 80–20 split and a 10-fold cross-validation model. The machine learning platform used in the present study was Azure Machine Learning, a cloud-based computing platform (Azure ML; Microsoft, Redmond, WA, USA). Nine 2-class classification algorithms were compared, including a neural network, averaged perceptron, a locally-deep support vector machine, logistic regression, a support vector machine, a Bayes point machine, a decision jungle, a boosted decision tree and a decision forest. After building the machine learning algorithms, the testing group was used to identify the algorithm with the greatest predictive power for fatty liver compared to the FLI.

### 2.4. Feature Selection

The clinical features included in our dataset were age, gender, height, weight, waist, BMI, systolic blood pressure (SBP), diastolic blood pressure (DBP), white blood cell count, red blood cell, hemoglobin, platelet, mean corpuscular volume, mean corpuscular hemoglobin, mean corpuscular hemoglobin concentration, red cell distribution width, aspartate aminotransferase, alanine aminotransferase (ALT), r-glutamyl transpeptidase (r-GT), total cholesterol, HDL, low-density lipoprotein, triglyceride (TG), creatinine, fasting serum glucose, Fibrosis-4 (FIB-4) score and FLI. Filter-based feature selection was used to rank each feature according to the calculated scores for each feature. Six feature scoring methods were used in the present study: Pearson’s correlation, mutual information, Kendall correlation, Spearman’s correlation, the chi-squared test and the Fisher score. After comparisons between ranking lists, we selected the 10 features most commonly used in clinical practice to build a prediction model.

### 2.5. FLI

The FLI was developed to predict hepatic steatosis in the general population by Giorgio Bedogni et al. [15] The FLI was calculated using the following equation:(e0.953 × loge (triglycerides) + 0.139 ×BMI + 0.718 × loge (r-GT) + 0.053 × waist circumference − 15.745)/(1 + e0.953 × loge (triglycerides) + 0.139 × BMI + 0.718 × loge (r-GT) + 0.053 × waist circumference − 15.745) × 100

### 2.6. Statistical Analysis

The chi-squared test was used to compare categorical variables, and the Mann–Whitney U test was used to compare continuous variables between groups. AUROC values were calculated using optimal cut-off values identified using the Youden index test. All statistical analyses were performed using SPSS version 22.0 (IBM Corp., Armonk, NY, USA), with 2-tailed *p* values < 0.05 indicating statistical significance.

## 3. Results

### 3.1. Clinical Characteristics of the Participants

Of the 12191 lean subjects, 741 (6.1%) had fatty liver. The training group comprised 8533 subjects (70%), of which 508 (6%) had fatty liver. The testing group comprised 3568 subjects (30%), of which 233 (6.4%) had fatty liver. The clinical characteristics of study subjects in the training group according to the presence or absence of fatty liver are shown in Table 1. Subjects with fatty liver were older, with a male predominance, compared to subjects without fatty liver. Significant differences in waist circumference, BMI, blood pressure, white blood cell count, serum hemoglobin, platelet count, serum glucose, serum biochemistry parameters, serum lipid profiles and FLI were observed between subjects with and without fatty liver. Histogram findings for both groups are shown in Figure 2. Accordingly, the figure shows that the distribution of the FLI is skewed to the right, with a slight difference between the two groups.

### 3.2. Feature Selection and Comparison of Classification Algorithms in the Training Set

Table 2 shows the 10 features selected by the six different scoring methods. The 10 highest scoring features were BMI, waist circumference, weight, age, serum TG, serum HDL, serum glucose, serum ALT, SBP and DBP. Table 3 shows the nine different classification algorithms of the machine learning models using the 10 selected features and all 27 features. The best predictive model was the two-class neural network using 10 features, with an AUROC value of 0.885, an accuracy of 0.816, a recall of 0.661, an F1 score of 0.72 and a precision of 0.791. (Figure 3).

### 3.3. Comparison of Machine Learning Models and the FLI Using the Testing Set

The FLI index was calculated in the testing group and compared to the machine learning model comprising a two-class neural network. The AUROC value was higher for the machine learning model using 10 selected features (above-mentioned; 0.868, 95% CI 0.841–0.894) compared to the FLI (0.852, 95% CI 0.824–0.881) and machine learning model using four selected features (same as factors in FLI including waist, BMI, r-GT and TG) (0.851, 95% CI 0.823–0.879; Figure 4). The optimal cut-off value, according to the Youden index test for the FLI, was 9, with a sensitivity of 82.4% and specificity of 74.9%.

## 4. Discussion

The present study is the first to use machine learning models to predict fatty liver in lean individuals. Our results demonstrate that a machine learning model had a slightly greater ability to discriminate between fatty liver and non-fatty liver in lean subjects from an Asian population compared to a conventional scoring system for predicting fatty liver (FLI). These machine learning algorithms may have utility in predicting fatty liver in regions where ultrasonography is not available and in analyzing pathophysiologic correlations with other clinical outcomes, such as cardiovascular events or mortality.

There is significant heterogeneity in the reported prevalence of fatty liver in lean individuals from previous studies due to the use of an upper limit of BMI ranging from 23 to 25 kg/m^2^ to define “lean” individuals in different populations. Two previous meta-analyses reported an overall prevalence of NAFLD in lean populations of 10.2% and 10.6% [16,17]. A lower prevalence of NAFLD was observed when lower BMI criteria were used and in populations based on patients attending health check-ups. The definition of “lean” in the present study was a BMI < 23 kg/m^2,^ and the prevalence of fatty liver in lean individuals was 6.1%. This prevalence of NAFLD in lean individuals was lower than reported by a similar study in a Taiwanese population conducted in 2019 which reported a prevalence of NAFLD of 18.5% in individuals with a BMI < 24 kg/m^2^ [8]. Other than differences in BMI cut-off values, criteria for diagnosing NAFLD using ultrasound may represent a further confounding factor as fatty liver was defined as moderate-to-severe fatty change on ultrasound in the present study. A similar prevalence (10.37%) of NAFLD in a non-obese population (BMI < 25 kg/m^2^) was reported from a study conducted in China which observed a higher prevalence of NAFLD in patients aged 50–59 years [18].

Differing factors are associated with NAFLD in lean and obese patients. The most common factors related to NAFLD in lean patients included BMI, waist circumference, serum triglyceride levels, serum HDL levels, DM, metabolic syndrome, blood pressure and serum liver enzyme levels [5]. Genetic factors are associated with the development of NAFLD in lean individuals, including patatin-like phospholipase domain-containing 3 (PNPLA3, rs738409 C > G) and transmembrane 6 superfamily member 2 (TM6SF2, rs58542926 C > T) [19,20]. The FLI contains four risk factors, including triglyceride, BMI, r-GT and waist circumference. Age is also an important risk of NAFLD in lean people. According to a meta-analysis, lean individuals with NAFLD were older than lean controls, with a mean difference of 2.87 years [21]. Sex differences have also been noted in NAFLD, such that a higher prevalence of NAFLD was found among males than among females. Liver outcomes in both sexes have been controversial, given that the effects of a dysmetabolic state might be greater in females than in males [22]. However, studies on the effects of sex on NAFLD in lean individuals have currently been limited. The present study used a cloud-based computing platform and filter-based selection to identify waist circumference, serum TG, BMI, weight, serum HDL, serum glucose, SBP, serum ALT, DBP and age as the 10 highest scoring factors. Although baseline characteristics showed male predominant in lean fatty liver, sex was not identified as an important feature in our model. These findings were similar to the results of previous studies [23,24].

AI has been used to predict NAFLD and classify the severity of liver fibrosis for many years [25]. A range of machine learning models and deep learning modules have been developed using large-scale electronic health records and images from clinical investigations, including histology, ultrasonography, computed tomography and magnetic resonance imaging [26]. Several public cloud-based platforms can be used to develop machine learning models, including Tensorflow (Google Brain Team, Menlo Park, CA, USA), WEKA (University of Waikato, Hamilton, New Zealand), the Orange Data Mining platform (Bioinformatics Lab, University of Ljubljana, Slovenia) and Azure Machine Learning (Microsoft, Redmond, WA, USA) [11]. According to a meta-analysis, AI-assisted ultrasonography had a higher area under the curve for the detection of NAFLD than an AI-assisted clinical data set (0.98 vs. 0.85) [9]. Several machine learning algorithms have been used to predict NAFLD, including extreme gradient boosting (XGBoost), neural network, support vector machine, logistic regression, decision forests and decision jungles. XGBoost is easy to interpret and has been shown to have good predictive utility in diagnosing NAFLD (AUROC, 0.882–0.931) in several studies [12,27,28]. While neural networks can address non-linear correlations and complex models, they require large computer resources and have a tendency to overfit. Meanwhile, support vector machines can handle high-dimensional data and non-linear correlations but have low efficacy due to their high computational costs. Logistic regression models are easy to train and interpret but cannot address non-linear problems. There was no available publication that used machine learning for the detection of lean NAFLD in the lean population. This is the first study to predict fatty liver in lean individuals using Azure Machine Learning which utilized nine different machine learning tools with a two-class neural network found to be the best machine learning algorithm (AUROC, 0.885). The predictive power is better in machine learning algorithms using 10 features than 27 features. The situation that performance was reduced by adding features may be explained by two possible reasons. First, when too many features are added to a machine learning model, it can lead to overfitting and influence the results from validation and test sets. Second, some features may not be useful in predicting outcomes or may be highly correlated with other features. When these features are added to the machine learning model, they can introduce noise and reduce the predictive value. After incorporating the neural network and the FLI into the testing group, we found that the neural network using 10 selected features had a slightly greater AUROC than did the FLI and neural network using four selected features (same as factors in FLI). This suggests the presence of non-linear correlations between these 10 features, which can be detected using certain specific machine learning algorithms. However, better information quality may be another possible explanation because the AUROCs are similar between the machine learning model using four features and FLI.

Scoring algorithms have been developed to predict hepatic steatosis. The FLI was developed in 2006 and had an accuracy for predicting steatosis of 0.84 [15]. Recently, the FLI has also been used to predict 10-year cardiovascular disease risks and mortality. [29] A study including a Korean population showed the individual who had a higher FLI (≥60) had a higher Framingham 10-year CVD risk (odds ratio 2.56). Another study from Korea showed that FLI could be a poor prognostic factor, particularly in the underweight population (BMI < 18.5 kg/m^2^). [30] Moreover, a study from China showed that a triglyceride and glucose index (TyG) developed to predict insulin resistance could also be used to identify NAFLD, although its AUROC was not particularly good (0.782). [31] Different cut-off values for the FLI for the prediction of fatty liver have been assessed in lean populations, with a previous study conducted in Taiwan demonstrating a cut-off value of 15 had the highest discriminant ability in a lean population and a further study conducted in Turkey demonstrated a cut-off value of 5.68 had the highest AUROC (0.748) for predicting NAFLD in lean females with polycystic ovary syndrome [8,32]. In the present study, the FLI had an AUROC value of 0.852 for predicting steatosis in lean individuals using a cut-off value of 9 identified using the Youden index test. We compared FLI cut-off values of 9 and 15 using a machine learning model in the testing group, with AUROC values of 0.784 for the FLI using a cut-off value of 9, 0.752 for the two-class neural network (scored probability > 0.5) and 0.722 for the FLI using a cut-off value of 15.

The present study had some limitations. First, we did not collect data on hepatitis B and C virus status or history of alcohol consumption or diabetes. The present study comprised subjects with fatty liver on ultrasound which may have included patients with NAFLD and alcoholic fatty disease. Although patients with viral hepatitis and alcohol liver disease may have been included in the present study, further studies are required to determine the predictive ability of machine learning models in detecting steatosis in populations with a single liver disease etiology. Metabolic-associated fatty liver disease (MAFLD) is not a suitable representation of our study subjects, given that our subjects do not satisfy the diagnostic criteria for MAFLD. Second, some clinical parameters were not included in the present study, such as uric acid, HbA1C, C-reactive protein and homeostasis model assessment for insulin resistance (HOMA-IR). These parameters have been shown to be associated with lean NAFLD [21,33]. Third, we did not collect histological data from liver biopsies or other non-invasive imaging techniques, such as the controlled attenuation parameter measured by FibroScan^®^ or magnetic resonance imaging-derived proton density fat fraction to confirm the degree of the steatosis. Our use of ultrasonography may have resulted in the underdiagnosis of hepatic steatosis in our patients relative to that using histology. Accordingly, the prevalence of fatty liver may have been underestimated in the present study. This would suggest that the machine learning model cannot be used to predict mild steatosis in the general population.

## 5. Conclusions

The present study describes the development of machine learning models to predict fatty liver in lean individuals, with a two-class neural network model found to have the greatest predictive ability. The AUROC value was slightly higher using the machine learning model than the FLI. In addition, the prevalence of fatty liver in lean individuals was 6.1%, and the optimal FLI cut-off value was 9 in the present study. These results can help clinicians promptly and accurately diagnose fatty liver disease in lean subjects considering that hepatic steatosis in lean populations can be easily overlooked or treated late. Further studies with larger sample sizes using other forms of clinical information and image are required to validate the utility of novel machine-learning models in predicting steatosis and fibrosis.

## Figures and Tables

**Figure 1 diagnostics-13-01407-f001:**
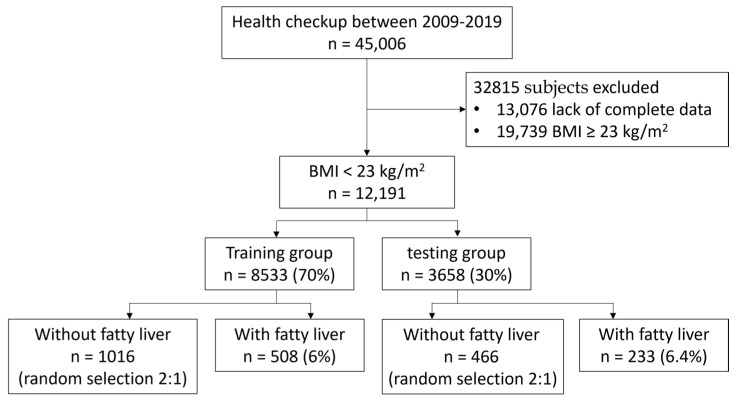
Participant flow chart for the training and testing groups in the present study.

**Figure 2 diagnostics-13-01407-f002:**
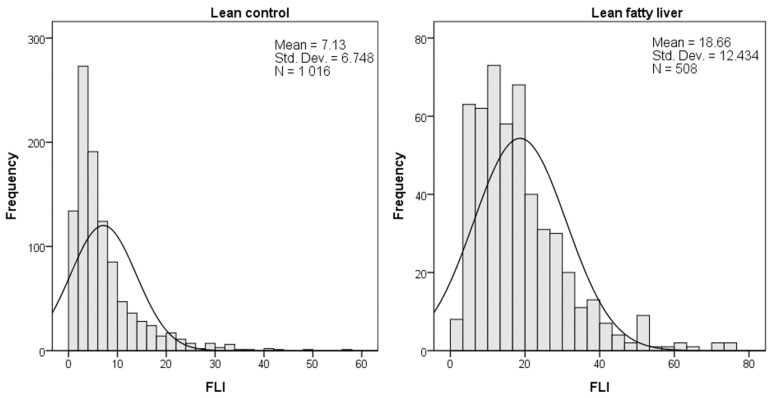
Histogram of fatty liver index (FLI) from lean fatty liver and control in the training group.

**Figure 3 diagnostics-13-01407-f003:**
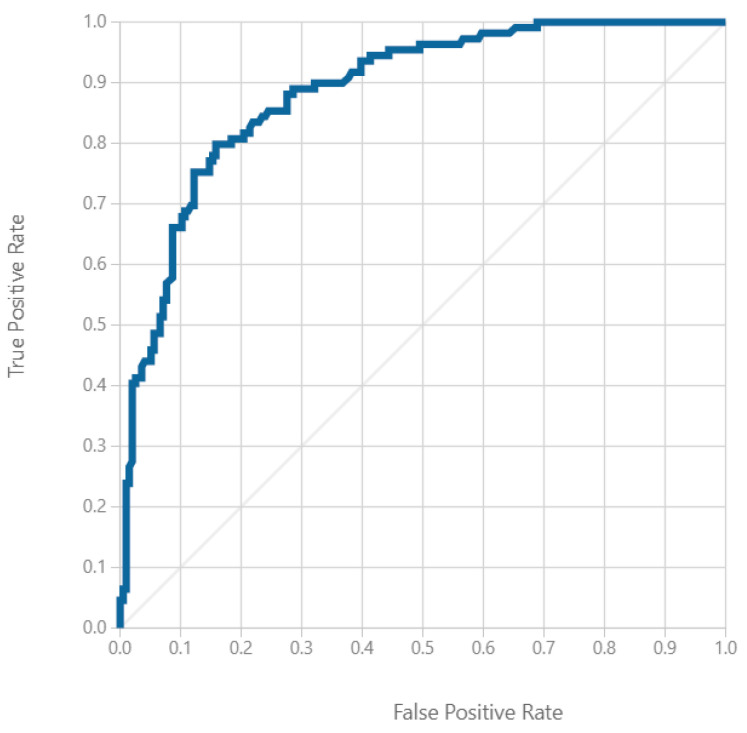
Area under the receiver operating characteristic curve for the machine learning model using a two-class neural network (AUROC value, 0.885).

**Figure 4 diagnostics-13-01407-f004:**
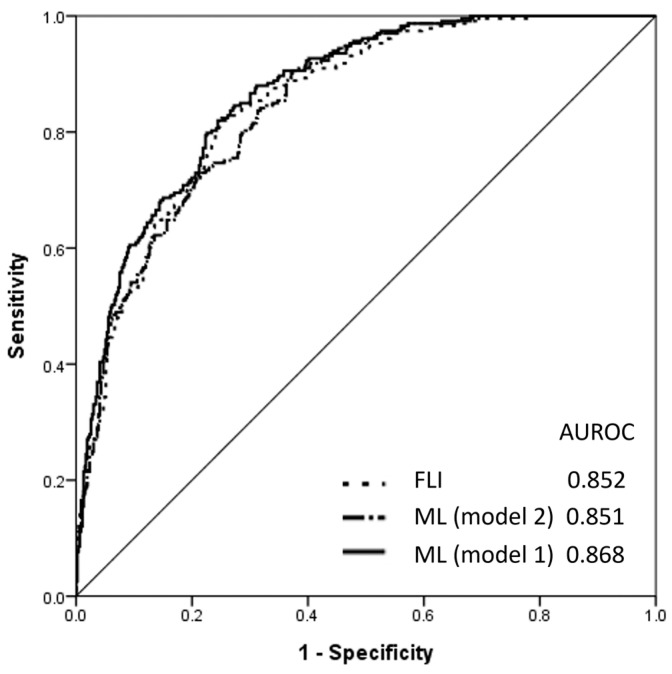
Area under the receiver operating characteristic (AUROC) curves of the machine learning (ML) model 1 comprising a two-class neural network using 10 selected features (body mass index, waist, weight, age, triglyceride (TG), high-density lipoprotein, glucose, alanine aminotransferase, systolic blood pressure and diastolic blood pressure), ML model 2 comprising a two-class neural network using four selected features (body mass index, waist, TG and r-glutamyl transpeptidase) and the fatty liver index (FLI) for predicting steatosis in lean individuals using the testing set.

**Table 1 diagnostics-13-01407-t001:** Clinical parameters in lean subjects with and without fatty liver.

	With Fatty Liver (*n* = 508)	Without Fatty Liver (*n* = 1016)	*p*-Value
Age (years)	52.69 ± 8.97	47.32 ± 11.17	<0.001
Gender, male	304 (59.8%)	438 (43.1%)	<0.001
Height (cm)	164.47 ± 8.11	163.52 ± 7.8	0.010
Weight (kg)	59.8 ± 6.31	56.04 ± 6.95	<0.001
Waist circumference (cm)	78.54 ± 5.1	73.14 ± 5.77	<0.001
SBP (mmHg)	127.24 ± 15.82	116.44 ± 14.57	<0.001
DBP (mmHg)	80.44 ± 9.5	74.42 ± 9.21	<0.001
BMI (kg/m^2^)	22.05 ± 0.78	20.9 ± 1.48	<0.001
WBC (×10^9^/L)	5.84 ± 1.49	5.18 ± 1.44	<0.001
RBC count (×10^9^/L)	4.76 ± 0.54	4.58 ± 0.48	<0.001
Hb (g/dL)	14.28 ± 1.36	13.73 ± 1.48	<0.001
MCV (fL)	42.57 ± 3.82	41.06 ± 4.1	<0.001
RBC volume (fL)	90.02 ± 7.46	90.1 ± 7.32	0.568
MCH (pg)	30.23 ± 2.95	30.14 ± 2.86	0.387
MCHC (g/dL)	33.54 ± 0.95	33.41 ± 0.88	0.001
Platelet (×10^3^/L)	226.59 ± 52.13	219.24 ± 49.68	0.009
RBC-RDW (%)	13.32 ± 1.04	13.51 ± 1.22	0.004
Glucose (mg/dL)	106.73 ± 31.21	91.48 ± 13.12	<0.001
AST (IU/L)	28.04 ± 15.42	23.03 ± 10.01	<0.001
ALT (IU/L)	31.36 ± 19.93	20.7 ± 12.26	<0.001
r-GT (U/L)	31.39 ± 56.94	17.68 ± 12.91	<0.001
Total cholesterol (mg/dL)	198.89 ± 37.48	190.62 ± 33.34	<0.001
HDL (mg/dL)	46.82 ± 10.66	57.06 ± 13.96	<0.001
LDL (mg/dL)	124.6 ± 34.5	115.23 ± 28.68	<0.001
Triglyceride (mg/dL)	138.14 ± 78.5	81.93 ± 45.67	<0.001
Cr (mg/dL)	0.78 ± 0.2	0.75 ± 0.18	0.001
FIB-4	1.26 ± 0.6	1.21 ± 0.63	0.014
FLI	18.66 ± 12.43	7.13 ± 6.75	<0.001

Abbreviations: SBP, systolic blood pressure; DBP, diastolic blood pressure; BMI, body mass index; WBC, white blood cell count; RBC, red blood cell; Hb, hemoglobin; MCV, mean corpuscular volume; MCH, mean corpuscular hemoglobin; MCHC, mean corpuscular hemoglobin concentration; RDW, red cell distribution width; AST, aspartate aminotransferase; ALT, alanine aminotransferase; r-GT, r-glutamyl transpeptidase; HDL, high-density lipoprotein; LDL, low-density lipoprotein; Cr, creatinine; FIB-4, fibrosis index based on four factors; FLI, fatty liver index.

**Table 2 diagnostics-13-01407-t002:** The 10 features selected by the six different scoring methods.

Method (Correlation Coefficient)	Scored Features										
	Steatosis	Waist	TG	BMI	HDL	Glucose	SBP	ALT	DBP	Weight	Age
Pearson correlation	1	0.417	0.412	0.388	0.349	0.325	0.321	0.313	0.292	0.254	0.235
	Steatosis	TG	BMI	Waist	rGT	ALT	Glucose	HDL	SBP	DBP	Age
Mutual information	1	0.105	0.097	0.094	0.089	0.086	0.078	0.067	0.055	0.045	0.039
	Steatosis	TG	Waist	rGT	ALT	BMI	Glucose	HDL	SBP	DBP	AST
Kendall correlation	1	0.364	0.348	0.341	0.334	0.325	0.310	0.297	0.264	0.243	0.212
	Steatosis	TG	Waist	rGT	ALT	BMI	Glucose	HDL	SBP	DBP	Weight
Spearman correlation	1	0.444	0.419	0.409	0.403	0.398	0.374	0.359	0.320	0.293	0.258
	Steatosis	TG	Waist	ALT	BMI	rGT	Glucose	HDL	SBP	DBP	AST
Chi-squared	1	316.830	272.800	264.647	260.506	255.513	236.273	197.071	164.514	136.655	117.880
	Steatosis	Waist	TG	BMI	HDL	Glucose	SBP	ALT	DBP	Weight	Age
Fisher Score	1	0.210	0.204	0.177	0.139	0.118	0.115	0.109	0.093	0.069	0.058

Abbreviations: TG, triglycerides; ALT, alanine aminotransferase; r-GT, r-glutamyl transpeptidase; BMI, body mass index; SBP, systolic blood pressure; DBP, diastolic blood pressure; AST, aspartate aminotransferase; HDL, high-density lipoprotein.

**Table 3 diagnostics-13-01407-t003:** Results from the nine machine learning models using the 10 selected features and all 27 features.

Model	Features	AUROC	Accuracy	Recall	F1 Score	Precision
Two-class neural network	10	0.885	0.816	0.661	0.72	0.791
	27	0.877	0.793	0.624	0.683	0.756
Two-class averaged perceptron	10	0.875	0.8	0.624	0.69	0.773
	27	0.874	0.79	0.615	0.677	0.753
Two-class locally-deep support vector machine	10	0.845	0.797	0.569	0.667	0.805
	27	0.833	0.77	0.505	0.611	0.775
Two-class logistic regression	10	0.867	0.793	0.596	0.674	0.774
	27	0.874	0.787	0.596	0.667	0.756
Two-class support vector machine	10	0.872	0.8	0.633	0.693	0.767
	27	0.866	0.793	0.642	0.69	0.745
Two-class Bayes point machine	10	0.86	0.77	0.505	0.611	0.775
	27	0.878	0.77	0.495	0.607	0.783
Two-class decision jungle	10	0.849	0.78	0.624	0.67	0.723
	27	0.851	0.784	0.578	0.656	0.759
Two-class boosted decision tree	10	0.859	0.764	0.624	0.654	0.687
	27	0.871	0.793	0.67	0.699	0.73
Two-class decision forest	10	0.852	0.784	0.56	0.649	0.772
	27	0.846	0.774	0.578	0.646	0.733

Abbreviations: AUROC, area under the receiver operating characteristic.

## Data Availability

Data are available on reasonable request.

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
