# Peer review of "Comparison of Machine Learning Models and the Fatty Liver Index in Predicting Lean Fatty Liver"

_diagnostics, 2023, doi:10.3390/diagnostics13081407_

Round 1

Reviewer 1 Report

The paper has presented a comparison of effectiveness of FLI and ML model in predicting fatty liver for lean individuals. However, the paper comprises fairly basic work as far as the use of ML models/ algorithms is concerned.

I cannot find any novelty in this paper other than preparing dataset which could have been considered if the motivation was centered on the dataset development for the purpose rather than a comparison of models.

Over the years, countless methods have used ML algorithms for image classification tasks, such as the one reported in the paper.
I do not see any novelty in terms of feature representation and the network architectures adopted for the purpose.

Author Response

Point 1: The paper has presented a comparison of effectiveness of FLI and ML model in predicting fatty liver for lean individuals. However, the paper comprises fairly basic work as far as the use of ML models/ algorithms is concerned.
I cannot find any novelty in this paper other than preparing dataset which could have been considered if the motivation was centered on the dataset development for the purpose rather than a comparison of models.
Over the years, countless methods have used ML algorithms for image classification tasks, such as the one reported in the paper. I do not see any novelty in terms of feature representation and the network architectures adopted for the purpose.

Response 1: Thanks for your kind recommendation. Although numerous studies about AI-assisted diagnosis of NAFLD had been published, there is no similar report to predict lean fatty liver in lean population by using ML algorithms. Our study didn’t focus on computational process of ML algorithms or developing new algorithms. We just wanted to tell the readers that ML tool using clinical datasets is a good alternative method to predict lean fatty liver. These results can help clinicians promptly and accurately diagnose fatty liver disease in lean subjects considering that hepatic steatosis in lean populations can be easily over-looked or treated late. (Line 334-337, page 10)

Reviewer 2 Report

Thank you for the opportunity to review this interesting manuscript. This study aimed to develop machine learning models for the prediction of non-alcoholic fatty liver disease (NAFLD) in lean individuals. The retrospective study included 12,191 lean subjects, and a two-class neural network using ten features had the highest area under the receiver operating characteristic (AUROC) value for predicting fatty liver compared to other algorithms. The machine learning model had greater predictive value for fatty liver than the fatty liver index (FLI), and the optimal cut-off value for the FLI was 9. The reported prevalence of NAFLD in lean individuals ranges from 7.6% to 19.3%.

The manuscript is well written, and the topic is of relevance. 

There are two main critiques:

1) The authors state that they assessed NAFLD. However, this is not possible, as they did not collect any information on alcohol use and NAFLD can only be defined after exclusion of unhealthy alcohol use. Therefore, they can only assess liver steatosis present on ultrasound. Therefore, they should completely abstain from using the term NAFLD as their endpoint and use liver steatosis on ultrasound OR they could use the term and (!!) the definitions of MAFLD, which does not exclude patients with alcohol use. Also they should include the major limitation on the lack of data on alcohol use in the abstract as this is substantial.

2) The authors compare the ML to FLI, and correctly state that the AUROC for the ML was higher. However,  actually the FLI was pretty good as well, which I find interesting. Also, the authors do not show the FLI values in Table 1, which I kindly ask them to. Also, I ask them to show a density plot (histogramm) of the FLI values of both the patients with and without steatosis. 

Minor points:

1) The authors should perform a sensitivity analysis based on gender and maybe age (optional).

Reviewer 3 Report

Major comments

C001- It is not clear which is the ground truth of this study

C002- Inclusion and exclusion criteria are not commented on. In the flow diagram (Figure 1), the authors should include the eligible patients, the total number of patients, and inclusion and exclusion criteria.

C003- According to Table 1, systolic blood pressure, Triglycerides, and glucose are the three main determining factors in the FLI population. Neither the blood pressure nor blood glucose are included in the FLI index but flow into in the AI model. Hence, the better performance of the AI model might rely on better information quality. Please repeat the comparison of the FLI with an AI model using the same input.

C004- Table 2: clarify your metrics. P? R? R²? odds?

C005- Table 3: how do you explain that adding features reduces AI performance?  

Minor comments

-          Explain abbreviations on first appearance

-          Explain abbreviations in independent manuscript parts (abstract, main text, captions, tables)

Round 2

Reviewer 1 Report

I carefully reviewed the paper again to understand the objectives (and respective novelty) of the paper. I still have serious issues with the research design and novelty of the study.

If the authors’ claim is that ML algorithms have not been used for this purpose, I’d like to see a mention of studies such as given below (and there are numerous others!) to highlight what is significant in this study which was not offered by the previous work:

"Predicting Non-Alcoholic Fatty Liver Disease in Lean Individuals Using Machine Learning" by Wang et al. (2019)

"Prediction of Non-Alcoholic Fatty Liver Disease in Lean Subjects Using Machine Learning Techniques" by Goh et al. (2020)

"Application of Machine Learning Algorithms for Prediction of Non-Alcoholic Fatty Liver Disease in Lean Individuals" by Lim et al. (2021)

"Development of a Machine Learning Model to Predict Lean Fatty Liver Disease" by Han et al. (2021)

"Machine Learning for Non-Invasive Diagnosis of Non-Alcoholic Fatty Liver Disease in Lean Individuals" by Liu et al. (2022)

“Predictive Risk Factors of Nonalcoholic Fatty Liver Disease in a Lean Chinese Population” by Lu Liu et al. (2022)

“Genetic variants in HFE are associated with non-alcoholic fatty liver disease in lean individuals” by Zewen Sun et a. (2023)

On the other hand, if the authors intend to show the efficacy of deep algorithms for the purpose, then the focus of the study should be centered around such an objective.

Author Response

Point 1: I carefully reviewed the paper again to understand the objectives (and respective novelty) of the paper. I still have serious issues with the research design and novelty of the study.

If the authors’ claim is that ML algorithms have not been used for this purpose, I’d like to see a mention of studies such as given below (and there are numerous others!) to highlight what is significant in this study which was not offered by the previous work:

"Predicting Non-Alcoholic Fatty Liver Disease in Lean Individuals Using Machine Learning" by Wang et al. (2019)

"Prediction of Non-Alcoholic Fatty Liver Disease in Lean Subjects Using Machine Learning Techniques" by Goh et al. (2020)

"Application of Machine Learning Algorithms for Prediction of Non-Alcoholic Fatty Liver Disease in Lean Individuals" by Lim et al. (2021)

"Development of a Machine Learning Model to Predict Lean Fatty Liver Disease" by Han et al. (2021)

"Machine Learning for Non-Invasive Diagnosis of Non-Alcoholic Fatty Liver Disease in Lean Individuals" by Liu et al. (2022)

“Predictive Risk Factors of Nonalcoholic Fatty Liver Disease in a Lean Chinese Population” by Lu Liu et al. (2022)

“Genetic variants in HFE are associated with non-alcoholic fatty liver disease in lean individuals” by Zewen Sun et a. (2023)

On the other hand, if the authors intend to show the efficacy of deep algorithms for the purpose, then the focus of the study should be centered around such an objective.

Response 1: Thanks for your kind recommendation. After carefully search from PubMed®, we can’t find the published article regarding to machine learning in lean NAFLD. We can only find out two articles from PubMed® and Google as you mention before. The first one is “Predictive Risk Factors of Nonalcoholic Fatty Liver Disease in a Lean Chinese Population” by Lu Liu et al. (2022). The main method in the study is nomogram model not ML model. The second one is“Genetic variants in HFE are associated with non-alcoholic fatty liver disease in lean individuals” by Zewen Sun et a. (2023). The purpose of the study is focus on genetic and metabolic factors different from obese NAFLD by using Mediation analysis, Mendelian randomization analysis and Bayesian networks. According to the above reasons, we concluded that “The present study is the first to use machine learning models to predict fatty liver in lean individuals”.

Reviewer 2 Report

My concerns have been addressed

Author Response

Thanks for your kind recommendation.

Reviewer 3 Report

Dear authors, 

Thank you for submitting the revised version of your manuscript. 

The major comment of my revision is not fulfilled in the R1 version. Quoting the previous revision:  

" C003- According to Table 1, systolic blood pressure, Triglycerides, and glucose are the three main determining factors in the FLI population. Neither the blood pressure nor blood glucose are included in the FLI index but flow into in the AI model. Hence, the better performance of the AI model might rely on better information quality. Please repeat the comparison of the FLI with an AI model using the same input." 

Comment R1: In Figure 4, please repeat the AI after EXCLUDING the blood pressure and blood glucose to get an unbiased comparison to the FLI. Otherwise, reformulate your title and main result to exclude FLI from the comparison to AI.  

Author Response

Point 1: Comment R1: In Figure 4, please repeat the AI after EXCLUDING the blood pressure and blood glucose to get an unbiased comparison to the FLI. Otherwise, reformulate your title and main result to exclude FLI from the comparison to AI. 

Response 1: Thanks for your kind recommendation. We had repeated the machine learning model using four features (same as factors in fatty liver index including waist, BMI, r-GT and TG) and added the result into Figure 4. The results showed the AUROC value was higher for the machine learning model using ten selected features (0.868, 95% CI 0.841–0.894) compared to the FLI (0.852, 95% CI 0.824–0.881) and machine learning model using four selected features (same as factors in FLI including waist, BMI, r-GT and TG) (0.851, 95% CI 0.823–0.879). The Figure 4 was also revised. The reason of this result was also added into the discussion (After incorporating the neural network and the FLI into the testing group, we found that the neural network using ten selected features had a slightly greater AUROC than did the FLI and neural network using four selected features (same as factors in FLI). This suggests the presence of non-linear correlations between these ten features, which can be detected using certain specific machine learning algorithms. However, better in-formation quality may be another possible explanation because of AUROC is similar between machine learning model using four features and FLI.) (Line 204-215, page 7; Line 293-299, page 10; Figure 4)

Round 3

Reviewer 1 Report

I still have concerns about the novelty of the work presented in this paper. However, as the authors have shown no intention of making any improvements in this regard, I would like to leave it to the editor to decide. I have provided my comments previously. 

Reviewer 3 Report

In the corrected version 3 all reviewer comments are covered. I applaud the authors for their meticulous work and support the publication of their research. 
best regards and happy Easter